# An MRI Radiomics Approach to Predict the Hypercoagulable Status of Gliomas

**DOI:** 10.3390/cancers16071289

**Published:** 2024-03-26

**Authors:** Zuzana Saidak, Adrien Laville, Simon Soudet, Marie-Antoinette Sevestre, Jean-Marc Constans, Antoine Galmiche

**Affiliations:** 1UR7516 CHIMERE, Université de Picardie Jules Verne, 80054 Amiens, France; saidak.zuzana@chu-amiens.fr (Z.S.); soudet.simon@chu-amiens.fr (S.S.); sevestre.marie-antoinette@chu-amiens.fr (M.-A.S.); constans.jean-marc@chu-amiens.fr (J.-M.C.); 2Service de Biochimie, Centre de Biologie Humaine, CHU Amiens, 80054 Amiens, France; 3INSERM UMR 1030, Gustave Roussy Cancer Campus, 94805 Villejuif, France; laville.adrien@chu-amiens.fr; 4Service de Radiothérapie, CHU Amiens, 80054 Amiens, France; 5Service de Médecine Vasculaire, CHU Amiens, 80054 Amiens, France; 6Service d’Imagerie Médicale, CHU Amiens, 80054 Amiens, France

**Keywords:** glioma, Glioblastoma Multiforme, magnetic resonance imaging (MRI), radiomics, venous thromboembolism, tissue factor

## Abstract

**Simple Summary:**

Venous thromboembolic events, such as pulmonary embolism and deep venous thrombosis, are frequent and potentially severe complications of gliomas. It is currently difficult to predict their occurrence, and systematic thromboprophylaxis is not recommended because of the associated risk of bleeding. The establishment of a local hypercoagulable state constitutes the biological rationale for these complications, with a key role played by the overexpression of tissue factor (TF). We examined the possible application of radiomics analyses of tumor MRI to predict the hypercoagulable status of gliomas. Using radiogenomics data from two independent glioma cohorts, we built a model that predicts the most hypercoagulable tumors with good performance. We discuss the potential of radiomics for the prediction of vascular complications and the study of coagulation in gliomas.

**Abstract:**

Venous thromboembolic events are frequent complications of Glioblastoma Multiforme (GBM) and low-grade gliomas (LGGs). The overexpression of tissue factor (TF) plays an essential role in the local hypercoagulable phenotype that underlies these complications. Our aim was to build an MRI radiomics model for the non-invasive exploration of the hypercoagulable status of LGG/GBM. Radiogenomics data from The Cancer Genome Atlas (TCGA) and REMBRANDT (Repository for molecular BRAin Neoplasia DaTa) cohorts were used. A logistic regression model (Radscore) was built in order to identify the top 20% TF-expressing tumors, considered to be at high thromboembolic risk. The most contributive MRI radiomics features from LGG/GBM linked to high TF were identified in TCGA using Least Absolute Shrinkage and Selection Operator (LASSO) regression. A logistic regression model was built, whose performance was analyzed with ROC in the TCGA/training and REMBRANDT/validation cohorts: AUC = 0.87 [CI_95_: 0.81–0.94, *p* < 0.0001] and AUC = 0.78 [CI_95_: 0.56–1.00, *p* = 0.02], respectively. In agreement with the key role of the coagulation cascade in gliomas, LGG patients with a high Radscore had lower overall and disease-free survival. The Radscore was linked to the presence of specific genomic alterations, the composition of the tumor coagulome and the tumor immune infiltrate. Our findings suggest that a non-invasive assessment of the hypercoagulable status of LGG/GBM is possible with MRI radiomics.

## 1. Introduction

Gliomas are invasive tumors of the central nervous system. High-grade gliomas, known as glioblastoma multiforme (GBM), represent the most frequent primary malignant tumor of the brain. Despite intensive therapeutic modalities, GBM has a nearly systematic fatal outcome marked by recurrence and a median overall survival of less than two years [1]. Venous thromboembolism (VTE) represents a great source of morbidity and mortality in cancer patients. GBM is considered to be the primary tumor type that is associated with the highest risk of VTE [2,3]. Approximately 20% of GBM patients encounter VTE and its potentially life-threatening complications, such as pulmonary embolism [4]. Low-grade gliomas (LGGs) are also associated with an increased risk of VTE, albeit at a slightly lower frequency than GBM [4]. While perioperative pharmacological thromboprophylaxis is proposed to GBM patients, anticoagulants are not systematically recommended for ambulatory patients with brain tumors, since the risk of bleeding counterbalances the benefit of VTE prophylaxis [4]. The most recent biological and clinical risk scores, such as the Khorana score, represent a significant advancement towards personalized VTE risk assessment in cancer patients, but their performance remains uncertain in the context of GBM [5]. The identification of new biomarkers that could predict the risk of VTE is a challenge for the optimization of medical care in LGG/GBM patients [6].

The establishment of a local hypercoagulable state constitutes the biological foundation for the high risk of VTE in cancer patients [7]. Cancer cells often overexpress tissue factor (TF), the pivotal activator of the coagulation cascade [8,9]. Among all primary tumors, GBM expresses the highest levels of the gene *F3*, encoding TF [10]. Meanwhile, LGG and GBM express low levels of the genes encoding the local activators of fibrinolysis, such as urokinase-type plasminogen activator (uPA) [10]. The strong and unopposed TF-dependent activation of coagulation therefore at least partially accounts for the formation of blood clots and the high risk of VTE in glioma patients [4,11]. Specific genomic alterations in tumors might regulate TF expression and therefore favor the occurrence of VTE [12]. Activation of the Epidermal Growth Factor Receptor (EGFR), through genomic amplifications or mutations, was reported to be positively correlated with *F3* expression in GBM [13]. A significant positive association between the amplification status of the EGFR locus in GBM and the presence of histologically detectable intra-tumoral thrombosis was suggested by Furuta et al. [14]; however, the link between VTE and the presence of EGFR amplifications in GBM nevertheless remains uncertain [15,16]. In addition to specific genomic alterations, chromosomal instability also activates innate immune pathways, potentially leading to *F3* expression in human tumors [17].

Importantly, the consequences of the activation of the coagulation cascade are not limited to the formation of thrombi in the lumen of the tumor vessels. The activated proteases of the coagulation cascade also exert direct effects on cancer cells and on the various cell types that constitute the tumor microenvironment (TME) [11]. In vitro studies support the protumoral role of the coagulation cascade activated by TF on the surface of glioma cells [18,19]. In GBM, activated thrombin was suggested to orchestrate the interaction between tumor cells, platelets, endothelial cells and immune cells in the TME [20]. As a consequence, the coagulation cascade is usually considered to play a detrimental role in LGG/GBM growth, dissemination and angiogenesis [20]. A recent experimental study directly addressed the role of TF in the response of GBM to radiotherapy, providing strong evidence for the role of TF in shaping the composition of the TME [21].

Magnetic Resonance Imaging (MRI) is the most frequent imaging modality used for the diagnosis of LGG/GBM. In addition to the essential morphological information that is immediately accessible to neuroradiologists, radiomics is a powerful, non-invasive strategy for the systematic assessment of tumor characteristics [22]. Extracting radiomic features from MRI images of LGG/GBM allows for the construction of prognostic models [23], or the non-invasive exploration of genomic alterations of tumors [24,25,26]. Other applications of radiomics also include the detection of epigenetic marks [27], the activated hypoxia pathway [28] and tumor infiltration by immune cells [29,30,31]. Consequently, radiomics holds great potential for the personalization of treatment decisions in neuro-oncology [32,33,34]. In the present study, we explored the possibility that an MRI-based radiomics model could identify the top 20% TF-expressing LGG/GBM tumors, considered to be hypercoagulable with high thromboembolic risk. To achieve this aim, we performed an integrative analysis, by combining transcriptomic and radiomic data available for LGG/GBM tumors in two independent cohorts, The Cancer Genome atlas (TCGA) and the REMBRANDT (REpository for molecular BRAin Neoplasia DaTa) cohorts, allowing for the integrative radiogenomics analysis of brain tumors [35,36,37,38,39].

## 2. Materials and Methods

### 2.1. Patient Information and Data Retrieval

The collection of all patient data from TCGA and REMBRANDT received Institutional Review Board approval. This work was carried out in accordance with the declaration of Helsinki, and informed consent was obtained from all patients [35,36,37]. Radiomic features previously extracted from the TCGA and REMBRANDT (REpository for molecular BRAin Neoplasia DaTa) LGG/GBM cohorts were retrieved on 1 November 2023 [38,39]. Details of the protocol on the technical aspects of preoperative MRI multi-sequence acquisition, processing and tumor segmentation can be found elsewhere [38,39]. Radiomics data were available for 243 patients in the TCGA cohort and 64 patients from REMBRANDT (*n* = 120 features). We retrieved transcriptomic data (*F3* expression) from cBioPortal for TCGA-LGG/GBM (RNA seq) or through the Gene Expression Omnibus GSE108476 (microarray) for REMBRANDT. We selected the tumors for which transcriptomic data (*F3* expression) and radiomics features were simultaneously available (*n* = 136 in TCGA, *n* = 39 in REMBRANDT). A detailed account of the patient information, data retrieval and handling of missing information can be found in the Appendix A and Methods. The clinical characteristics of the patients in the two cohorts are summarized in Table 1. No significant differences were apparent upon a direct comparison of the basic clinical characteristics of the two cohorts, as shown in Appendix A.

### 2.2. Model Construction and Validation

The TCGA-LGG/GBM cohort was used to build the model. The REMBRANDT cohort was used as an independent validation cohort. TCGA-LGG/GBM tumors were randomly split (7:3) into training (*n* = 95) and testing sets (*n* = 41). Co-correlated features were eliminated (r > 0.85). The training dataset was used to identify the most important features linked to high *F3* expression (top 20%, *F3*^high^) using LASSO regression analysis (alpha = 1). Nested cross validation was used to identify the most stable features (10-fold nested crossed validation, repeated 10 times) [40]. The key features retained for the model were selected based on the lambda parameter and stability analyses, and a logistic regression model (Radscore) was built using the top 7 most stable and contributive features. A list of the features and their coefficients is presented in Appendix A. Details on the model construction and the R packages used can be found in the Appendix A. The performance of the model was assessed with Receiver Operating Characteristic (ROC) analysis in TCGA and in the independent validation cohort (REMBRANDT).

### 2.3. Tumor Analyses and Transcriptional Signatures

Gene Set Enrichment Analysis (GSEA) was performed with the Java GSEA desktop application (https://www.gsea-msigdb.org/gsea/index.jsp (accessed on 15 December 2023)) [41]. We queried the Hallmarks gene sets to compute the enrichment of specific gene sets in tumors stratified according to their Radscore. The 70-gene mRNA expression signature CIN70 (chromosomal instability) was previously reported [42] and recently applied to evaluate the chromosomal instability in LGG/GBM [43]. Aneuploidy scores were retrieved from Thorsson et al. [44]. The aneuploidy score reflects the sum of amplified or deleted chromosomal arms, with arms considered altered if the mean fraction altered is >80% [44]. The immune score has been previously described [45]. Briefly, it is based on the analysis of the transcriptional enrichment of a set of 141 genes reflecting the presence of immune cells in human tumors. The CIBERSORTx algorithm (https://cibersortx.stanford.edu (accessed on 5 December 2023)) was used to quantify the levels of 22 cell subsets using the validated leukocyte gene signature matrix LM22 [46,47].

### 2.4. Descriptive Statistical Analyses

Comparisons of numeric data were performed using Wilcoxon–Mann–Whitney or Student’s *t* test. Comparisons of categorical data were performed with a Chi-squared test. Kaplan–Meier analyses and the log-rank test were used to compare the overall survival (OS) and disease-free survival (DFS) between groups. Details on the performance analyses and the list of R packages used are provided in the Appendix A. *p* < 0.05 was set as the threshold for significance. False Discovery Rate (FDR) correction was applied as indicated. We used R version 4.3.1 (https://www.r-project.org). 

## 3. Results

### 3.1. Construction and Validation of an MRI Radiomics Model That Reflects the Hypercoagulable Status of Human Gliomas

To build and validate a radiomics model linked to the hypercoagulable phenotype of LGG/GBM, we used two cohorts allowing for integrative radiogenomics studies of primary brain tumors, i.e., TCGA and REMBRANDT [38,39]. The main characteristics of the patients in the two cohorts are presented in Table 1 and Appendix A.

We noted that around 20% of LGG/GBM tumors in TCGA express high levels of *F3* (*F3*^high^) (Appendix A). The prevalence of VTE is estimated to be around 20% in LGG/GBM patients [4]. Given the key role of TF in cancer-associated thrombosis [7,8,9,10], we hypothesized that *F3*^high^ tumors might represent hypercoagulable tumors associated with a high risk of VTE. Our aim was therefore to build a model to identify the 20% of tumors with the highest *F3* expression. A flow chart summarizing the construction of the model is presented in Figure 1.

LASSO regression is well suited for building sparse models with a reduced number of features. We applied LASSO regression to the TCGA/training set, in order to identify the most important radiomics features linked to the hypercoagulable status of LGG/GBM (*F3*^high^). The optimal number of parameters was identified through lambda analysis to be between 7 and 16 features (Figure 2A). The seven most stable and important radiomics features, ranked according to their importance, are shown in Figure 2B. These seven features, which reflect tumor morphology and tumor texture, were used to construct a logistic regression model (Radscore) (features and coefficients are listed in Appendix A). The performance of the model was tested with ROC analysis, giving an AUC of 0.87 [95% CI: 0.81–0.94, *p* < 0.0001] in the TCGA cohort (Figure 2C). Independent validation of its performance in the REMBRANDT cohort gave an AUC of 0.78 [0.56–1.00, *p* = 0.02] (Figure 2C).

In order to further assess the performance of the model, an optimal cut-off value was calculated using the MaxProdSpSe method (maximizing the product of specificity and sensitivity) (cut-off = 2.06 in TCGA). The performances of the model for the detection of hypercoagulable LGG/GBM in both cohorts are summarized in Table 2 (accuracy = 0.79 [CI_95_: 0.71–0.85] in TCGA and 0.74 [CI_95_: 0.58–0.87] in REMBRANDT, respectively). The confusion matrices for TCGA and REMBRANDT are provided in Appendix A. Overall, we concluded that the Radscore has acceptable performance in predicting the hypercoagulable status in LGG/GBM.

Both the TCGA and REMBRANDT cohorts are enriched in LGG compared to GBM (78.7% LGG in TCGA, and 82.0% LGG in REMBRANDT). We separately examined the performance of the model in LGG and GBM in TCGA, giving an AUC of 0.88 [95% CI: 0.80–0.95, *p* < 0.0001] and AUC of 0.80 [95% CI: 0.53–1.00, *p* = 0.03] for LGG and GBM, respectively (Appendix A). It was not possible to perform this subset analysis in REMBRANDT due to the limited number of tumors (*n* = 7 GBM).

### 3.2. Clinical and Prognostic Significance of the Radscore

To further analyze the potential use of the Radscore, we examined the clinical significance of the model. We used Kaplan–Meier analyses to examine the OS and DFS of patients with tumors identified as hypercoagulable by the Radscore (top 20% highest score). Given the different clinical courses of LGG and GBM, we analyzed their survival separately. The median survival times (OS, DFS) and hazard ratios (HR) were determined in Radscore^high^ vs. Radscore^low^ tumors for the different cohorts/tumors analyzed (TCGA/LGG, TCGA/GBM, REMBRANDT/LGG, REMBRANDT/GBM), showing consistently lower median survival and increased HR for Radscore^high^ tumors (Appendix A). As shown in Figure 3A, high Radscore values were associated with a significantly lower OS (*p* = 0.0034) and DFS (*p* = 0.0012) in LGG patients in TCGA. A similar tendency was observed in REMBRANDT (*p* = 0.12) (Figure 3B). A decrease in OS was observed in TCGA/GBM with a high Radscore, but it did not reach statistical significance (*p* = 0.057) (Figure 3A).

We further examined the pathological significance of the Radscore of LGG/GBM by addressing the link between the Radscore and some of the key genomic characteristics previously reported to be of prognostic importance (Figure 4). Radscore^High^ tumors were enriched in GBM tumors compared to LGG (37.9% of GBM were Radscore^high^ vs. 15.9% of LGG, *p* = 0.0059 with Chi^2^) and IDH1 wild-type status (87.0% vs. 29.0%, *p* < 0.000001). The copy number alterations (amplifications, gains) of the EGFR locus were also significantly more frequent in Radscore^high^ tumors (77.8% vs. 36.8%, *p* = 0.000158) (Figure 4A). Given the reported links between tumor genomic instability, the activation of the innate immune system and the induction of a procoagulant transcriptional program including *F3* [17], we also examined the link between the Radscore and tumor aneuploidy using scores retrieved from Thorsson et al. [44] and Carter et al. [42] (Figure 4B). Radscore^High^ tumors had higher aneuploidy scores and CIN70 scores compared to the others (*p* = 6.902 ×10^−6^ and *p* = 0.0015, respectively, using Wilcoxon–Mann–Whitney) (Figure 4B). A subset analysis performed in TCGA, where we separately examined LGG and GBM, suggested that the link between aneuploidy/chromosomal instability and the Radscore was significant in LGG, but not in GBM (Appendix A).

### 3.3. Tumor Microenvironmental Characteristics of Gliomas Stratified According to Radscore

Next, we wanted to address the link between the Radscore and the TME. As expected, we observed a significant difference in the mRNA levels of *F3* between Radscore^high^ and Radscore^low^ tumors in TCGA and REMBRANDT cohorts (Figure 5A). We also performed a Gene Set Enrichment Analysis (GSEA) for a larger and unbiased comparison of the transcriptional profiles of TCGA-LGG/GBM stratified according to the Radscore. Interestingly, the hallmark gene set «COAGULATION» was shown to be significantly upregulated in Radscore^high^ tumors with a Normalized Enrichment Score (NES) = 2.0 (Figure 5B). A leading-edge analysis identified 31 coagulation-related genes overexpressed in Radscore^high^ LGG/GBM (Appendix A). We concluded that the model reflects broader characteristics of the tumor coagulome than just the *F3* expression that was used to build it.

Finally, we examined the immune infiltration characteristics of the tumors stratified according to the Radscore with two independent approaches: (i) The IMMUNE score, previously established and validated in primary brain tumors [45]; (ii) the CIBERSORTx algorithm [46,47]. Radscore^high^ tumors had a significantly higher IMMUNE score (*p* = 2.037 × 10^−5^), suggestive of higher immune infiltration/activity (Figure 5C). A CIBERSORTx analysis was performed on TCGA to assess the infiltration of 22 immune cell populations, expressed as relative fractions and as absolute scores (Appendix A). A statistical analysis of the relative fractions, using the Chi^2^ test, found no statistically significant difference between Radscore^high^ and Radscore^low^ tumors (Appendix A). However, the absolute score analysis showed that Radscore^high^ tumors were significantly more infiltrated with M0 macrophages and regulatory T cells (Tregs): a 4.5-fold higher level of M0 macrophages (*p* = 0.0017) and a 2.4-fold higher level of Tregs (*p* = 0.0044) (Appendix A). A Pearson correlation analysis showed a significant correlation between the Radscore and absolute infiltration levels of these two cell types (Pearson r = 0.32, *p* = 0.0001 for M0 macrophages; r = 0.37, *p* = 7.95 × 10^−6^ for Tregs) (Appendix A, Figure 5D).

## 4. Discussion

Despite the high incidence of VTE and its complications in LGG/GBM patients, systematic thromboprophylaxis is currently not indicated. The identification of glioma patients at the highest risk of VTE is an important goal from the perspective of defining a population with a favorable risk/benefit ratio for thromboprophylaxis [4]. Using a combination of features that reflect tumor morphology and texture, we built a Radscore that reflects the expression of *F3*, a key actor of the coagulation cascade, in gliomas [18,19,20]. The Radscore had prognostic value and appeared to be linked to some of the key molecular alterations found in gliomas (IDH1 mutational status, genomic EGFR amplification, presence of tumor aneuploidy and chromosomal instability) in accordance with the important role played by TF in gliomagenesis, and the regulatory role played by some of these alterations in *F3* expression [12,13,14,17]. Our study suggests for the first time that MRI radiomics may be helpful for the identification of hypercoagulable tumors at high risk of VTE. The radiomics approach that we describe is a novel and promising strategy for exploring the thromboembolic risk of individual LGG/GBM tumors. It might be particularly valuable for better patient stratification according to the thromboembolic risk. Considering the high negative predictive value of the Radscore, it might be used to identify glioma patients who are less susceptible to VTE. Considering the non-invasive nature of MRI, the possibility of repeating the analysis of the tumor coagulome several times, for example, in a therapeutic context or upon tumor recurrence, might also be an important advantage.

Our study has a number of limitations. The validation cohort (REMBRANDT) that we used here included a small number of patients (*n* = 39 gliomas, with only seven patients with GBM). This limited number of patients obviously explains why we observed broad confidence intervals in the validation analyses of the Radscore performance. The small number of patients, especially regarding GBM, also prevented us from carrying out detailed subset analyses of the Radscore’s performance in GBM. With these limitations in mind, we chose not to use any data augmentation procedures, given the risk of introducing bias associated with these procedures. Importantly, the two cohorts that we used in this study also used different protocols for gene expression analysis (RNA seq in TCGA vs. microarray in REMBRANDT). Despite the normalization procedures that we applied in this study, we cannot exclude the existence of a bias that may have influenced our results and the performance of the Radscore. Another limitation of this study is the fact that we could not directly address the risk of VTE or its complications in LGG/GBM patients. Indeed, TCGA and REMBRANDT provide access to a limited set of clinical data, with no useful biological parameters linked to coagulation, such as blood levels of D-dimers. Nevertheless, we based our approach on the strong rationale provided by the observation that GBM tumors have high expression levels of *F3* and few or no activators of fibrinolysis, leading to the unopposed activation of TF-driven coagulation [10]. We constructed our radiomics model under these premises. While other genes in addition to *F3* may be associated with the risk of VTE, their contribution remains unclear [4,6]. Importantly, we observed that Radscore^high^ tumors tend to express high levels of a large array of genes related to coagulation, strongly suggesting that the Radscore reflects broader characteristics of the tumor coagulome than just the *F3* expression that was used to build it. At this stage, our study provides a proof of principle that MRI radiomics could be used to predict the hypercoagulable status of gliomas, rather than an established model with a fixed list of features. More studies are required to link tumor radiomics with the occurrence of VTE and its complications. We propose that the optimal personalized prediction of the risk of VTE in LGG/GBM will require prospective integrative studies combining tumor radiomics with an extended set of clinical and biological parameters [6].

Finally, the impact of the hypercoagulable status of LGG/GBM extends beyond thromboembolic risk. A number of experimental studies suggest that the coagulation cascade plays a detrimental role in LGG/GBM growth, dissemination and angiogenesis via the interaction of coagulation proteases with cognate receptors on the surface of cancer cells and cells of the TME [18,19,20]. The activation of the coagulation cascade is increasingly recognized as a determinant of the cellular composition and functional state of the TME through its effects on fibroblasts, endothelial cells and myeloid and immune cells [11]. Using the same digital cytometric analyses that were recently used to unveil the importance of the immune cell infiltrate in LGG/GBM [48,49,50], we observed that a high Radscore is associated with the presence of increased levels of non-polarized M0 macrophages (4.5-fold increase compared to Radscore^low^) and Tregs (2.4-fold increase compared to Radscore^low^). Given the complex regulation and role of the tumor immune ecosystem in LGG/GBM, we can only speculate about the clinical relevance of these differences in immune infiltrate [48,49,50]. Nevertheless, our observations on macrophages seem reminiscent of the conclusions of a recent study exploring the role of TF in GBM treated with radiotherapy. In *F3*-positive, fibrin/fibronectin complex-rich regions of irradiated GBM, Jeon et al. reported a significant increase in the number of tumor-associated macrophages [21]. The findings of this study and others [11] highlight the existence of a link between the coagulation cascade and the recruitment/activation of tumor-associated macrophages in LGG/GBM. Importantly, the study of Jeon et al. also suggests that a hypercoagulable tumor status might be of particular importance in a therapeutic context. Upon surgical bleeding/radiotherapy, the systemic activation of coagulation may act in synergy with the local hypercoagulable state created by the tumor itself to further modulate tumor growth or shape the tumor immune microenvironment. In this setting, predicting the hypercoagulable status of a tumor using a radiomics approach, such as the one we propose here, could be useful for patient prognosis and for predicting tumor response to therapy.

## 5. Conclusions

Our findings suggest that a non-invasive assessment of the hypercoagulable status of LGG/GBM is possible with MRI radiomics. We propose that a radiomics approach that captures the hypercoagulable phenotype of LGG/GBM may be of interest for a more personalized prediction of the risk of VTE. It might also be of great help in exploring the dynamic regulation of the tumor coagulome and addressing the role of coagulation in the TME and as a determinant of tumor response to therapy.

## Figures and Tables

**Figure 1 cancers-16-01289-f001:**
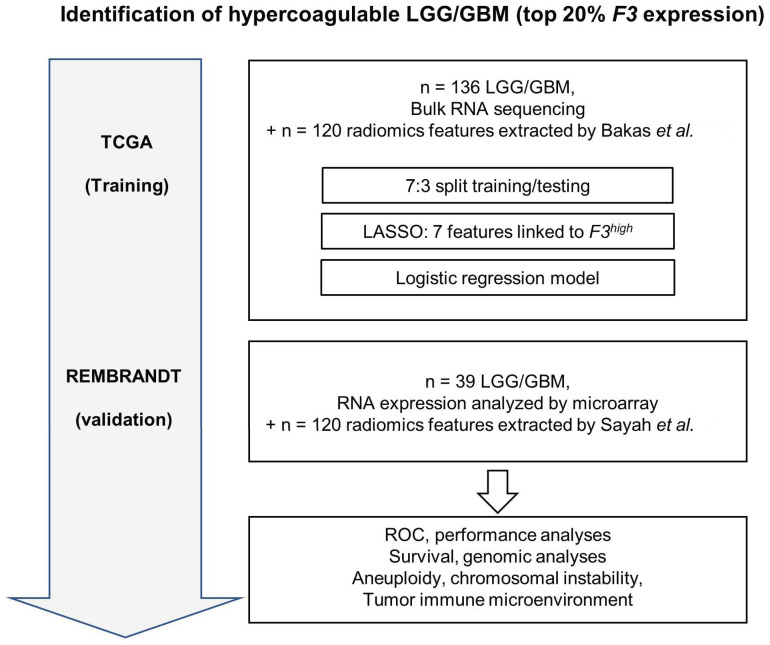
Construction of an MRI radiomics model that predicts the hypercoagulable status of human gliomas. The two cohorts used in this study were TCGA (*n* = 136 gliomas), used for model construction, and the REMBRANDT cohort (*n* = 39), used for independent validation [38,39].

**Figure 2 cancers-16-01289-f002:**
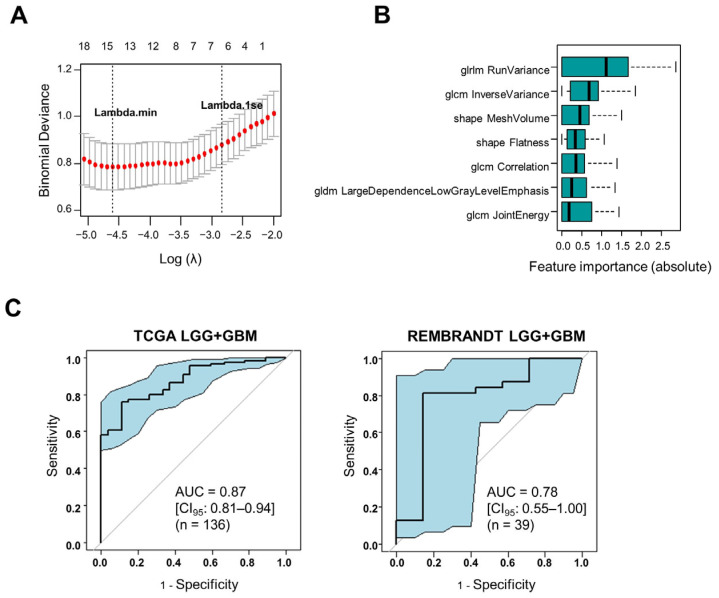
Construction and validation of an MRI radiomics model that predicts the hypercoagulable status of human gliomas. (**A**) Lambda analysis (LASSO) to determine the optimal number of features for the model. (**B**) The 7 most stable features linked to *F3*^high^, retained in the logistic regression model (Radscore), organized by their importance. (**C**) Performance analysis (ROC) of the Radscore in identifying the top 20% *F3*^high^ tumors in the TCGA (training) and REMBRANDT (validation) cohorts. The shaded areas indicate CI_95_.

**Figure 3 cancers-16-01289-f003:**
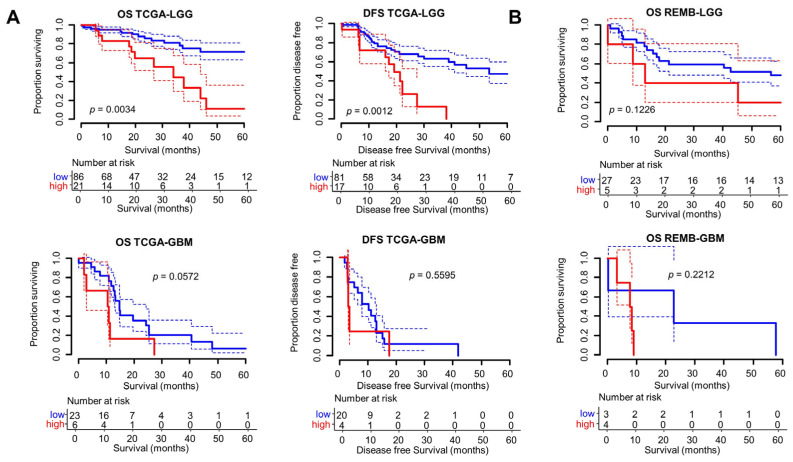
Survival analyses of patients with LGG/GBM according to the Radscore. Kaplan–Meier analyses of OS and DFS of LGG and GBM patients in TCGA (**A**) and REMBRANDT (**B**) cohorts, comparing the survival of patients with the top 20% highest Radscore (red) vs. the others (blue) (*p* values calculated using the log rank test). The dotted lines indicate the CI_80_.

**Figure 4 cancers-16-01289-f004:**
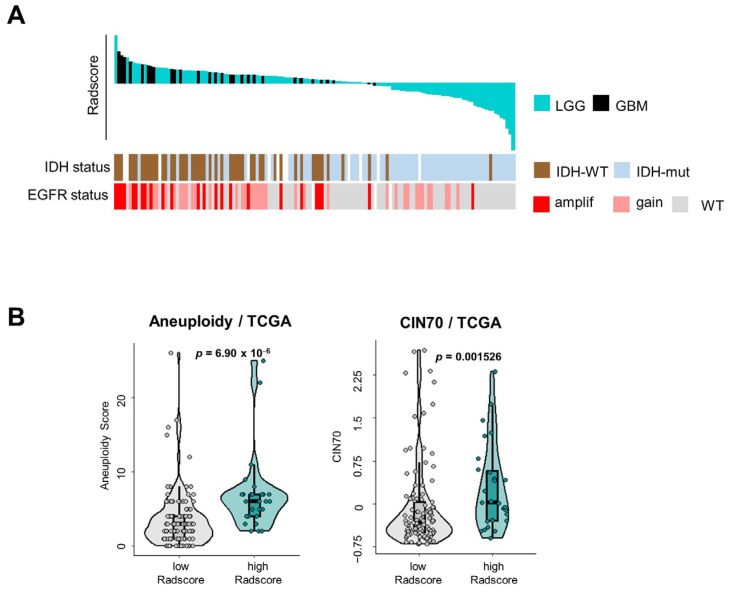
Molecular characteristics of gliomas stratified according to the Radscore. (**A**) IDH mutational status and EGFR copy number alterations in TCGA tumors ranked by decreasing Radscore, in LGG (turquoise) and GBM (black). Chi-squared test was used for statistical comparisons. (**B**) Comparison of the aneuploidy and CIN70 scores in TCGA gliomas stratified according to the Radscore (Wilcoxon-Mann–Whitney).

**Figure 5 cancers-16-01289-f005:**
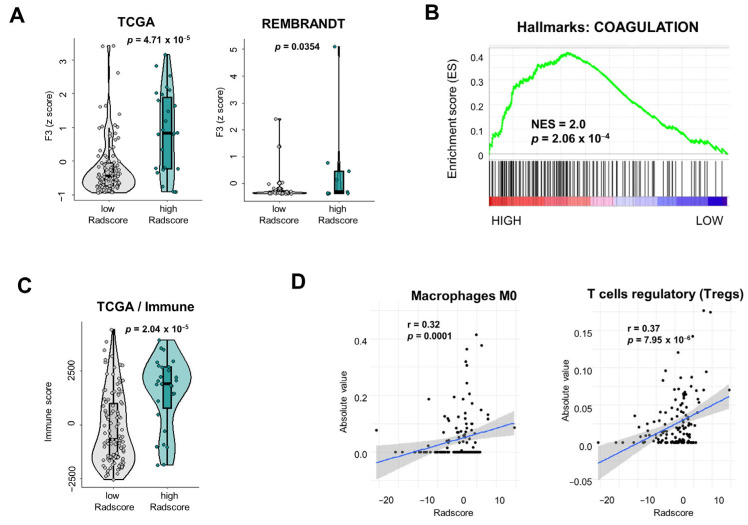
Tumor microenvironmental characteristics of gliomas stratified according to the Radscore. (**A**) *F3* mRNA expression levels in Radscore^high^ vs. Radscore^low^ gliomas in TCGA and REMBRANDT. (**B**) Gene Set Enrichment Analysis (GSEA) analysis for the «COAGULATION» Hallmark gene set. (**C**) IMMUNE score was calculated for TCGA samples according to the Radscore. (**D**) Pearson correlation analysis between the Radscore and the absolute infiltration levels of M0 macrophages and Tregs in TCGA.

**Table 1 cancers-16-01289-t001:** Clinical characteristics of LGG/GBM patients in TCGA and REMBRANDT cohorts. WT: wild type; NA: not available.

Clinical Characteristics	TCGA (*n* = 136)	REMBRANDT (*n* = 39)
Age (years): mean, range	48.8 (20–79)	48 (18–87)
Sex (male, female, NA)	69 (51%), 67 (49%), 0 (0%)	21 (54%), 14 (36%), 4 (10%)
Histology:		
Astrocytoma	79 (58%)	27 (69%)
Oligodendroglioma	28 (21%)	5 (13%)
Glioblastoma	29 (21%)	7 (18%)
Grade: II, III, IV	48 (35%), 58 (43%), 29 (21%)	20 (51%), 12 (31%), 7 (18%)
IDH1 status:		
Mutated, WT, NA	75 (55%), 51 (38%), 10 (7%)	0, 0, 39 (100%)

**Table 2 cancers-16-01289-t002:** Model performance *.

Parameters	TCGA	REMBRANDT
AUC	0.87 [0.81–0.94]	0.78 [0.56–1.00]
Sensitivity	0.89 [0.71–0.98]	0.86 [0.42–1.00]
Specificity	0.76 [0.67–0.84]	0.72 [0.53–0.86]
Positive predictive value	0.48 [0.34–0.63]	0.40 [0.16–0.68]
Negative predictive value	0.97 [0.90–0.99]	0.96 [0.79–1.00]
Accuracy	0.79 [0.71–0.85]	0.74 [0.58–0.87]

* performance values are presented by CI_95_ in both cohorts for a Radscore cut-off value of 2.06.

## Data Availability

All data used in this study are freely available through cBioPortal (TCGA) and Gene Expression Omnibus (GSE108476), and as supplementary material in Bakas et al. [38] and Sayah et al. [39]. The codes used for the construction of the Radscore are available as Appendix A.

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
