# Peer review of "An MRI Radiomics Approach to Predict the Hypercoagulable Status of Gliomas"

_cancers, 2024, doi:10.3390/cancers16071289_

Round 1

Reviewer 1 Report

Comments and Suggestions for Authors

The authors show that high expression of tissue factor in glioma is associated with specific MRI-features that correlate with reduced survival, increased oncogenic activity, a reduced immune response and overall upregulation of coagulation pathways. A high Radscore seemed particularly relevant in patients with low-grade glioma which exhibited a significantly lower overall survival. Unfortunately, direct proof of hypercoagulability in patients with a high radscore such as increased fibrin in glioma tissue or increased D-Dimers in glioma patients was not included. Nevertheless, these very exciting data provide proof of principal that activation of specific genetic programs in cancer such as the clotting system are detectable by imaging techniques and have the potential to provide valuable information with regard to thromboembolic complications and overall outcome if validated properly.

Minor comment:

The impact of a high radscore in LGG compared to GBM is quite striking but not further discussed. The authors should explain why they think this is and what it means in terms of risk stratification.

Author Response

Reviewer #1

The authors show that high expression of tissue factor in glioma is associated with specific MRI-features that correlate with reduced survival, increased oncogenic activity, a reduced immune response and overall upregulation of coagulation pathways. A high Radscore seemed particularly relevant in patients with low-grade glioma which exhibited a significantly lower overall survival. Unfortunately, direct proof of hypercoagulability in patients with a high radscore such as increased fibrin in glioma tissue or increased D-Dimers in glioma patients was not included. Nevertheless, these very exciting data provide proof of principal that activation of specific genetic programs in cancer such as the clotting system are detectable by imaging techniques and have the potential to provide valuable information with regard to thromboembolic complications and overall outcome if validated properly.

Minor comment: The impact of a high radscore in LGG compared to GBM is quite striking but not further discussed. The authors should explain why they think this is and what it means in terms of risk stratification.

We thank the reviewer #1 for these positive comments on our study. We agree with the inherent limitations of our study, especially regarding the lack of a direct proof of hypercoagulability. To the best of our knowledge, there are no radiomics cohorts that would provide access to MRI and fibrin / D-dimers levels in LGG/GBM patients. As the reviewer states, the study is presently more of a proof of principle. Ideally, a solid validation would be needed through a prospective study specifically addressing the link between radiomics and the hypercoagulable status of LGG/GBM, reflected by biomarker analyses (D-dimers and fibrin). We rephrased the 2nd paragraph of the discussion (p. 10 of the revised mansucript) to more clearly discuss this point.

As requested, we have inserted a comment in the discussion to describe the opportunity that radiomics could offer for risk stratification. The relatively good NPV of the Radscore could for example be used as a way to enrich clinical trials examining thromboprophylaxis in LGG/GBM with people at risk of VTE (2nd paragraph of the revised discussion, p10 of the manuscript). We think that it is however difficult to compare the clinical significance of the Radscore in LGG and GBM, considering the great differences in the oncological course of the two types of tumors.

We are grateful to reviewer #1 for helping us to propose a more criticial / clearer discussion of our findings.

Reviewer 2 Report

Comments and Suggestions for Authors

Authors describe a logisitc regression prediciton model on the hypercoagulable state in gliomas based on tissue factor expression and radiogenomics analyses of tumor MRI by making use of open data.
The manuscript is well written and results are comprehensibly presented. Radscore model feature importance, LGG and GBM separately, as well as relation to known genomic characteristics have been investigated.
Methods mainly should be refined in terms of reproducibility. Since this work is based on open data only, consider code availability and provide source code via a repository with indications in methods and next to the data availability statement!
Furthermore, package version numbers are missing.

Author Response

Reviewer #2

Authors describe a logisitc regression prediciton model on the hypercoagulable state in gliomas based on tissue factor expression and radiogenomics analyses of tumor MRI by making use of open data. The manuscript is well written and results are comprehensibly presented. Radscore model feature importance, LGG and GBM separately, as well as relation to known genomic characteristics have been investigated.

Methods mainly should be refined in terms of reproducibility. Since this work is based on open data only, consider code availability and provide source code via a repository with indications in methods and next to the data availability statement ! Furthermore, package version numbers are missing.

               We thank reviewer #2 for the positive comments on the writting and the presentation of our study. We agree with the reviewer that reproducibility is a major point. We have made an effort to mention the origin of the genomic data in the main manuscript, that was originally only mentioned in Suppl. materials (see p3 of the revised manuscript). We have also included the codes that were used to split TCGA cohort, perform nested cross-validation, lambda analysis and Radscore calculation. The codes can be found in the last two pages of the « Supplementary materials and methods » file, as mentioned in the « Data Availability Statement » (p11 of the revised manuscript). Package versions are now given in the relevant section of the « Suppl. Materials and Methods » (nestedcv version 0.7.4, glmnet version 4.1-8, caret version 6.0-94, OptimalCutpoints version 1.1-5, Survival 3.5-8, ggplot2 version 3.4.4, epiR version 2.0.68).

Reviewer 3 Report

Comments and Suggestions for Authors

The manuscript by Saidak et al. assesses associations between MRI-based symptoms and the F3 transcript levels that might be considered as characteristic for hypercoagulable state in Glioblastoma Multiforme (GBM) and Low-Grade Gliomas (LGG).

MRI-based symptoms were selected via poorly described DL-based image segmentation algorithm followed by the selection of features via the LASSO regression algorithm.

The F3 transcript levels were estimated via two different methods – RNASeq in TCGA and microarray in REMBRANDT cohorts

Then authors constructed the simple ML-based model utilising TCGA and REMBRANDT cohorts for training and testing, without estimating uncertainties of performance indicators using cross-validation or other methods.

Several critical limitations might bias the performance of the constructed method when changing the method of transcript levels estimation.

While the work suggests that F3 transcript levels are not the ultimate marker of the GBM tumours, the authors also tested the associations of several other indicators, including IDH, Tumor microenvironmental characteristics and others.

The manuscript is logical in general; it nevertheless requires a number of improvements before publication.

Major issues

[1]

The authors received good AUC scores for the TCGA cohort but failed to replicate the performance of the model using the REMBRANDT cohort, achieving a wide uncertainty interval for AUCs estimated.

Several possible reasons on the observed limited performance of the model with the REMBRANDT cohort must be provided within the “Discussion” section.

[1a]

Since validation of the constructed algorithm was completed on the REMBRANDT dataset which was based on the different method of F3 transcript levels estimation, the F3 transcript levels were estimated in training and validation groups using two intrinsically different methods, which might bias the performance of the algorithm while changing the method of transcript levels detection.

This intrinsic bias might result in not achieving high AUCs in the REMBRANDT cohort.

Please discuss this critical limitation of the results in a special subsection “Limitations” within the “Discussion” section, considering the possible consequences.

[1b]

Another reason of not achieving high AUCs in the REMBRANDT cohort is the suboptimal method of transcript level estimations chosen states that the maximal value was retained in cases of multiple rows available for the same gene (“F3 expression analysis” subsection in the  Suppl materials)

To deal with this possible limitation of the study, please clarify (In the Suppl materials) the reasons to use the maximal value and not to use the integral expression.

Please also discuss this limitation in the relevant subsection of the “Discussion” section.

[1c]

The Histograms of F3 expression in TCGA and REMBRANDT depicted at the Fig S1 suggest that Microarray is poorly suitable for transcript level estimation.

Due to differences between microarrays and RNA-Seq, the suggested 20% cutoff for the F3 levels might be ineffective for microarray-based estimations of expression.

Please discuss the reasons of choosing the suggested 20% cutoff as universal one and possible limitations of such a decision in the relevant subsection of the “Discussion” section.

[1d]

In the “Abstract” section, please directly reflect the uncertainty of the AUC indicators in a proper way.

[2]

The authors declared that small number of patients that were included is a critical limitation of this study.

In the relevant subsection of the “Discussion” section, please discuss the reasons of not using the dataset augmentation procedures.

[3]

In the “Discussion” section, the authors must discuss top 7 features extracted by LASSO regression in the context of their meanings for diagnostics of GBM LGG tumours.

[4]

The Figure 1 does not provide a clear linear representation of the developed pipeline and was not refereed properly in the relevant subsections of the manuscript.

[4a]

The representation of the whole workflow must be improved greatly to ensure clear representation of the analysis pipeline, identifying clear correspondence between the text details and the Figure 1 elements.

For clarity, it is possible to use the Y-axis as a global time axis discriminating the timeline of pipeline steps and substeps performed.

[4b]

Similar, the present description of the pipelines applied for (a) MRI data and (b) for transcriptomic data processing also is not illustrative and misses some clarity and integral view. Please improve via extending the Fig. 1.

Both MRI-based and expression-based pipelines must be reflected on the Fig, with differences between train and test samples.

The illustrated pipelines must be quoted within the relevant subsections of the manuscript extensively, identifying clear correspondences between the text details and the Pipeline Figure elements.

[5]

While the general description of the methods is good, some important details are missing in the text.

[5a]

The detailed description of the automated volumetric segmentation of MRIs, which was used to identify the tumour necrotic core, edema and other features, must be provided in the Suppl materials. The references for the applied code also must be mentioned within the text of the Suplement.

[5b]

“120 radiomic features were extracted using Pyradiomics” – In the text of the Supplement, please clarify this episode mentioned in the “Radiomics features, handling of missing information” subsection of the Supplementary Materials

[5c]

In the relevant subsection, please provide the reasoning to select top 20% F3-expressing tumours as ones with high expression

[5d]

In the relevant subsections of the Suppl Materials, please provide the ranges of possible values that can be achieved via calculations of Radscore, aneuploidy scores and the CIN70 (Chromosomal Instability) scores.

[6]

The data reporting must be improved extensively.

[6a]

All figures with boxplots (like Comparison of the aneuploidy and CIN70 scores on the Fig 4B or F3 mRNA expression levels on the Fig 5A) must be visualised in a form of Raincloud plots with jittering, the Median, IQR boxes and CI95 intervals depicted for each quantitative characteristic.

To achieve this, it is possible to use the freeware JASP https://jasp-stats.org/ or another freeware/software capable of drawing Raincloud plots with jittering

[6b]

Similar, demographic and clinical characteristics must be visualized as Supplementary Raincloud plots with jittering, the Median, IQR boxes and CI95 intervals visualized for each quantitative characteristic like age, between control and affected samples.

[6c]

In the Table 2, the authors have provided the performances of the model in both cohorts without the uncertainty intervals assessed by cross-validation (or another relevant technique applicable for variance estimation).

Please depict the Sensitivity, Specificity, PPV, NPV, Accuracy and F1-score results as means and SDs or medians and IQRs.

[6d]

In the Table 2, please also provide the AUC values for the model tested, with the uncertainty intervals assessed by cross-validation or another relevant technique.

[6e]

On the Fig. 3 panels, the survival analysis lines must be presented with true CI95 (or CI90, or other) confidence bands.

For help, please refer to Coviello (https://www.graphpad.com/support/faq/confidence-intervals-vs-confidence-bands-for-survival-curves/) or other relevant source of information .

[6f]

Similar, the ROC curves depicted on the Fig 2C panels must be provided with the corresponding CI95 bands estimated via cross-validation or another relevant technique.

[6g]

The depiction of the ROC curves on the Fig S2 must be improved in a similar way.

[6h]

The full results of the Cybersortx algorithm must be provided in the supplement and compared statistically.

Please check cell numbers of each type for classes compared by the Chi-squared test or Fischer Exact test for 2xN contingency tables with the CI95 ranges identified and visualised. Such testing might help to reveal disproportions in the sampled cell numbers for specific cell types.

To do so, please see online freeware “Epitools” (https://epitools.ausvet.com.au/chisq), which might depict CI95 intervals for cell numbers, or another statistical software/freeware capable to complete such a task.

The estimation of proportions for sampled cell numbers must be done for all types of cells simultaneously for a RxC matrix. Please do not use 2x2 designs! Please use RxC design.

[6i]

L257 - CIBERSORT or CIBERSORTx? Please clarify

[7]

The authors provided the information on all R tools applied in this work. Nevertheless, it would be great if the readers would be provided with the full list of code sources used by the authors.

All tools applied by the authors must be mentioned in the way similar as the “List of R packages used” table and referenced within the relevant sections of the text extensively.

Comments on the Quality of English Language

[8]

Language issues

The language is comprehensible but missing some specific points.

Please check the text for any unclear clauses that are not obvious in a direct way.

Several examples are below.

L48 - “a frequent complication” of what? – Please clarify

LL52-53 “a hypercoagulable state” global? local? – Please clarify

L72 - “was suggested to have a positive effect on F3 expression in GBM” – Please clarify

L73 – “A significant association” – positive? negative? – Please clarify

L73 - “the EGFR amplification status of GBM” – Please clarify

L92 – “radiomics is emerging as” – Present Continuous? Please check the Tense

LL94-95 –“allows for example for the construction” – Please check te word order.

L164 – “overexpress” overexpresses?

Please check the manuscript extensively for similar incomplete clauses.

Author Response

Reviewer #3

The manuscript by Saidak et al. assesses associations between MRI-based symptoms and the F3 transcript levels that might be considered as characteristic for hypercoagulable state in Glioblastoma Multiforme (GBM) and Low-Grade Gliomas (LGG).

MRI-based symptoms were selected via poorly described DL-based image segmentation algorithm followed by the selection of features via the LASSO regression algorithm.

The F3 transcript levels were estimated via two different methods – RNASeq in TCGA and microarray in REMBRANDT cohorts

Then authors constructed the simple ML-based model utilising TCGA and REMBRANDT cohorts for training and testing, without estimating uncertainties of performance indicators using cross-validation or other methods.

Several critical limitations might bias the performance of the constructed method when changing the method of transcript levels estimation.

While the work suggests that F3 transcript levels are not the ultimate marker of the GBM tumours, the authors also tested the associations of several other indicators, including IDH, Tumor microenvironmental characteristics and others.

The manuscript is logical in general; it nevertheless requires a number of improvements before publication.

We thank Reviewer #3 for critically examining our study and we appreciate the comments that our research is overall presented in a logical way. Our answers to each detailed comment or criticism can be found below.

[1] The authors received good AUC scores for the TCGA cohort but failed to replicate the performance of the model using the REMBRANDT cohort, achieving a wide uncertainty interval for AUCs estimated. Several possible reasons on the observed limited performance of the model with the REMBRANDT cohort must be provided within the “Discussion” section.

We agree with reviewer #3 that the Radscore performs slightly less well in the REMBRANDT cohort, but we do not agree with the statement that « we failed to replicate the performance of the model in REMBRANDT ». We believe that an AUC=0.78 denotes an acceptable performance.

As the reviewer later mentions, the REMBRANDT cohort is much smaller than TCGA, with less data and also some technical limitations. A few possible biases might explain why the performances of the Radscore are lower in REMBRANDT. Nevertheless, we believe that our study makes an adequate use of the data obtained by Sayah et al. [39], produced with exactly the same pipeline and made available by the same renowned team that had previously extracted radiomics features from LGG/GBM of TCGA [38]. The reference [39] explains how REMBRANDT can be used to validate radiogenomics studies and meta-analyses in conjunction with TCGA, exactly as we did in the present study.

Below, we discuss in detail the points that we changed in the revised version of the manuscript in order to meet the requests of reviewer #3.

[1a] Since validation of the constructed algorithm was completed on the REMBRANDT dataset which was based on the different method of F3 transcript levels estimation, the F3 transcript levels were estimated in training and validation groups using two intrinsically different methods, which might bias the performance of the algorithm while changing the method of transcript levels detection. This intrinsic bias might result in not achieving high AUCs in the REMBRANDT cohort. Please discuss this critical limitation of the results in a special subsection “Limitations” within the “Discussion” section, considering the possible consequences.

We agree with the reviewer #3 that the use of different protocols to analyze gene expression might have introduced bias. In the revised version of the manuscript, we have introduced a line in the section of the discussion devoted to this limitation of the study (2nd paragraph of the discussion, p10 of the revised manuscript).

[1b] Another reason of not achieving high AUCs in the REMBRANDT cohort is the suboptimal method of transcript level estimations chosen states that the maximal value was retained in cases of multiple rows available for the same gene (“F3 expression analysis” subsection in the  Suppl materials). To deal with this possible limitation of the study, please clarify (In the Suppl materials) the reasons to use the maximal value and not to use the integral expression. Please also discuss this limitation in the relevant subsection of the “Discussion” section.

As the reviewer rightly mentions, microarray analyses rely on the use of a small number of probes for each gene. Conflicting information can be obtained, for example if a specific splicing variant is only detected with one probe. We chose to retain in each case the maximal expression value, considering that we would analyse the most expressed mRNA species. An integral analysis or a calculation of the average value could be legitimate, but it also has its own potential bias. We introduced a few sentences regarding this choice in the revised version of the Supplementary Materials and methods (paragraph « F3 expression analysis »).

[1c] The Histograms of F3 expression in TCGA and REMBRANDT depicted at the Fig S1 suggest that Microarray is poorly suitable for transcript level estimation. Due to differences between microarrays and RNA-Seq, the suggested 20% cutoff for the F3 levels might be ineffective for microarray-based estimations of expression. Please discuss the reasons of choosing the suggested 20% cutoff as universal one and possible limitations of such a decision in the relevant subsection of the “Discussion” section.

               We apologize for not having introduced the rationale for the choice of a 20% cut-off with enough detail. The choice is based on our initial analysis of the distribution of F3 in the TCGA (shown in Suppl. Fig. 1) combined with the common medical knowledge that approximately 20% of patients with LGG/GBM encounter VTE and its complications. We agree with the reviewer that the use of two different techniques for gene expression analysis might have introduced a bias, and that adapting the cut-off depending on the cohort analysed would have improved the performances of the Radscore in REMBRANDT. We chose to stick to the 20% cut-off for consistency, thinking that a change would have been criticised as data dredging.

[1d] In the “Abstract” section, please directly reflect the uncertainty of the AUC indicators in a proper way.

We agree with the reviewer that this important point was missing. It is now introduced in the revised version of the abstract.

[2] The authors declared that small number of patients that were included is a critical limitation of this study. In the relevant subsection of the “Discussion” section, please discuss the reasons of not using the dataset augmentation procedures.

We agree with the reviewer that a data augmentation approach would be enticing, but we do not master this type of approach. Meanwhile, data augmentation is challenging and not entirely without risk, such as the introduction of biases in the augmented data.

               Also, as briefly mentioned in our response to reviewer #1, we want to attract the attention of the reviewer again to an essential point: our study provides a proof of principle that radiomics accurately reflects the hypercoagulable status of gliomas, but we do not claim that our Radscore is ready-to-use in the clinics. Expanding the number of patients would not address the major criticism raised by reviewer #1, that we indirectly examined the hypercoagulable status. We believe that the definitive validation of a radiomics model will only result from prospective studies that include biological parameters, such as D-dimers, that are not available in TCGA or REMBRANDT.

               Considering that dataset augmentation procedures would not address this essential limitation of our study, and given our lack of expertise on the topic, we chose not to discuss it. Instead, in the revised discussion we insist on the need to perform a prospective validation with an extended array of biological parameters (see p10 of the revised manuscript).

[3] In the “Discussion” section, the authors must discuss top 7 features extracted by LASSO regression in the context of their meanings for diagnostics of GBM LGG tumours.

The seven features that were retained (glrlm_RunVariance, glcm_InverseVariance, shape_MeshVolume, shape_Flatness, glcm_Correlation, gldm_LargeDependenceLowGrayLevelEmphasis, glcm_JointEnergy) reflect tumor morphology and tumor texture. These parameters are occasionally identified in other radiomics studies, but it is difficult to discuss their contribution to the diagnosis of LGG/GBM, since radiomics is hardly used for this application.

Once again, a prospective study matching glioma radiomics with a direct characterization of their hypercoagulable status would be required to validate the approach. We believe that a heavy discussion of each parameter is premature, given that slightly different radiomics features might be identified in subsequent prospective studies including a direct biological assessment of the tumor hypercoagulable state.

[4] The Figure 1 does not provide a clear linear representation of the developed pipeline and was not refereed properly in the relevant subsections of the manuscript.

[4a] The representation of the whole workflow must be improved greatly to ensure clear representation of the analysis pipeline, identifying clear correspondence between the text details and the Figure 1 elements. For clarity, it is possible to use the Y-axis as a global time axis discriminating the timeline of pipeline steps and substeps performed.

[4b] Similar, the present description of the pipelines applied for (a) MRI data and (b) for transcriptomic data processing also is not illustrative and misses some clarity and integral view. Please improve via extending the Fig. 1. Both MRI-based and expression-based pipelines must be reflected on the Fig, with differences between train and test samples.

The illustrated pipelines must be quoted within the relevant subsections of the manuscript extensively, identifying clear correspondences between the text details and the Pipeline Figure elements.

               A revised Fig. 1 is proposed, where the composition of each cohort is more clearly detailed. A Y-axis arrow is introduced. The analytic pipeline is also more detailed, with a mention of the gene expression technique used in each cohort.

Nevertheless, we would like to underline that MRI processing and radiomics feature extraction were produced by other investigators, in two previous studies that we have cited (references [38] and [39] of the manuscript). Introducing these steps in Fig. 1 would falsely give the impression that we claim to have done these steps. We chose to refer to the original studies in Figure 1 for clarity.

[5] While the general description of the methods is good, some important details are missing in the text.

[5a] The detailed description of the automated volumetric segmentation of MRIs, which was used to identify the tumour necrotic core, edema and other features, must be provided in the Suppl materials. The references for the applied code also must be mentioned within the text of the Suplement.

[5b] “120 radiomic features were extracted using Pyradiomics” – In the text of the Supplement, please clarify this episode mentioned in the “Radiomics features, handling of missing information” subsection of the Supplementary Materials

               We agree with the reviewer that segmentation / feature extraction are two key steps in radiomics studies. Nevertheless, we would like to attract the attention of the reviewer to the fact that we did not perform these analytic steps ourselves. We used the data that had been extracted from automatically segmented tumors analyzed by American colleagues working at the « Center for Biomedical Image Computing and Analytics », at the University of Pennsylvania (USA). The two research papers that describe how the radiomics features were generated are cited in our manuscript as ref. [38] and [39].

In the revised version of the manuscript, we significantly reorganized the Supplementary materials and methods section devoted to the description of radiomics features. We now mention the use of GLISTRboost, a computer-aided segmentation tool used by the American neuroradiologists that produced the radiomics data [38,39]. We mention that the sub-regions of the brain were identified as necrotic core, edema, non-enhancing tumor and enhancing tumor, Gray Matter, White Matter, and Cerebrospinal Fluid. A Board-Certified radiologist verified and refined the segmented labels. As reported in detail in the corresponding research papers, 120 features were extracted volumetrically and consist of i) intensity, ii) volumetric, iii) morphologic, iv) histogram-based and v) textural parameters, including features based on wavelets, GLCM (Gray Level Co-occurrence Matrix), Gray Level Dependence Matrix (GLDM), Gray-Level Run-Length Matrix (GLRLM), Gray-Level Size Zone Matrix (GLSZM), and Neighborhood Gray-Tone Difference Matrix (NGTDM).

Once again, we think that giving more information would falsely give the impression that we claim to have done these steps. Instead, we consistently refer to the work of the two research papers [38] and [39], that were published in the journal « Scientific Data », an open-access journal devoted to the description of datasets and advances in sharing and reuse of scientific data. All codes used, all images and segmentation labels are freely available through TCIA, Github, or as supplementary material to Ref. [38] and [39]. We hope that these elements will clarify the source of the data that we used, and at the same time convince the reviewer that they are freely available.

[5c] In the relevant subsection, please provide the reasoning to select top 20% F3-expressing tumours as ones with high expression

               See our response to the comment [1c].

[5d] In the relevant subsections of the Suppl Materials, please provide the ranges of possible values that can be achieved via calculations of Radscore, aneuploidy scores and the CIN70 (Chromosomal Instability) scores.

We have inserted the requested data in the suppl. materials.

[6] The data reporting must be improved extensively.

[6a] All figures with boxplots (like Comparison of the aneuploidy and CIN70 scores on the Fig 4B or F3 mRNA expression levels on the Fig 5A) must be visualised in a form of Raincloud plots with jittering, the Median, IQR boxes and CI95 intervals depicted for each quantitative characteristic.

To achieve this, it is possible to use the freeware JASP https://jasp-stats.org/ or another freeware/software capable of drawing Raincloud plots with jittering

               We have replaced the box plots of Fig. 4B and 5A with violin plots with jittering.

[6b] Similar, demographic and clinical characteristics must be visualized as Supplementary Raincloud plots with jittering, the Median, IQR boxes and CI95 intervals visualized for each quantitative characteristic like age, between control and affected samples.

               We sincerely apologize to the reviewer, but we would like to ask her/him to reconsider this request. We have never before seen such an advanced graphic presentation of basic demographic and clinical data. We believe that this analysis would not add any useful information to the present study, that is not already present in Table 1. We hope that the reviewer will understand our point of view.

[6c] In the Table 2, the authors have provided the performances of the model in both cohorts without the uncertainty intervals assessed by cross-validation (or another relevant technique applicable for variance estimation). Please depict the Sensitivity, Specificity, PPV, NPV, Accuracy and F1-score results as means and SDs or medians and IQRs.

[6d] In the Table 2, please also provide the AUC values for the model tested, with the uncertainty intervals assessed by cross-validation or another relevant technique.

               We have calculated the requested uncertainty intervals with the R package epiR 2.0.68 and they are inserted in Table 2. The AUC is also inserted in Table 2 with the associated CI95 (p6 of the revised manuscript).

 [6e] On the Fig. 3 panels, the survival analysis lines must be presented with true CI95 (or CI90, or other) confidence bands.

For help, please refer to Coviello (https://www.graphpad.com/support/faq/confidence-intervals-vs-confidence-bands-for-survival-curves/) or other relevant source of information.

               We did not manage to introduce a visual representation of the CI95 in our survival analyses. Our version of GraphPad Prism (version 5) likely does not provide this option. We apologize for this, and nevertheless ask the reviewer to consider that the great majority of studies that are published do not include CI lines in survival analyses. In order to enrich the statistical analyses presented at this level, we provide instead the median OS/DFS of Radscorehigh vs Radscorelow LGG / GBM with the corresponding hazard ratios with 95% CI. In the revised manuscript, the data are inserted in a new Table S2.

[6f] Similar, the ROC curves depicted on the Fig 2C panels must be provided with the corresponding CI95 bands estimated via cross-validation or another relevant technique.

[6g] The depiction of the ROC curves on the Fig S2 must be improved in a similar way.

               We did not manage to produce the requested data in the limited amount of time that was awarded to us. Here again, we ask the reviewer to reconsider this request, considering that the great majority of radiomics studies do not include this uncertainty visualization. 

[6h] The full results of the Cybersortx algorithm must be provided in the supplement and compared statistically. Please check cell numbers of each type for classes compared by the Chi-squared test or Fischer Exact test for 2xN contingency tables with the CI95 ranges identified and visualised. Such testing might help to reveal disproportions in the sampled cell numbers for specific cell types.

To do so, please see online freeware “Epitools” (https://epitools.ausvet.com.au/chisq), which might depict CI95 intervals for cell numbers, or another statistical software/freeware capable to complete such a task.

The estimation of proportions for sampled cell numbers must be done for all types of cells simultaneously for a RxC matrix. Please do not use 2x2 designs! Please use RxC design.

[6i] L257 - CIBERSORT or CIBERSORTx? Please clarify

               We apologize to the reviewer for misspelling the name of the CIBERSORTx algorithm. We have re-analyzed the data with a 2xN Chi2 test, and lost statistical significance. This is not surprising, considering that the cell fraction that we considered the most interesting (M0 macrophages) are present in small numbers in LGG/GBM. The Fig. 5D and former Table S3 were consequently removed in the revised version of the manuscript. 

               Instead, in the revised version of the manuscript, we now present a correlation analysis between the Radscore and the density of each immune cell infiltrate (CIBERSORTx). The new Table S3 identifies M0 macrophages as the cell population that is best correlated with the Radscore in LGG/GBM (Pearson r=0.26, p=0.0024). Instead of a box plot, the new Fig. 5D illustrates this correlation. The former Table S3 (now Table S4) lists all Pearson correlation coefficients for each immune cell population. We are grateful to reviewer #3 for attracting our attention to the statistical analysis of CIBERSORTx results.

[7] The authors provided the information on all R tools applied in this work. Nevertheless, it would be great if the readers would be provided with the full list of code sources used by the authors.

All tools applied by the authors must be mentioned in the way similar as the “List of R packages used” table and referenced within the relevant sections of the text extensively.

               We agree with the reviewer #3 regarding this important point. The codes that we used for the identification of the radiomics features and the Radscore itself are now included in a specific section in the revised « Suppl. Materials and methods ».

[8] The language is comprehensible but missing some specific points.

Please check the text for any unclear clauses that are not obvious in a direct way.

Several examples are below.

L48 - “a frequent complication” of what? – Please clarify

LL52-53 “a hypercoagulable state” global? local? – Please clarify

L72 - “was suggested to have a positive effect on F3 expression in GBM” – Please clarify

L73 – “A significant association” – positive? negative? – Please clarify

L73 - “the EGFR amplification status of GBM” – Please clarify

L92 – “radiomics is emerging as” – Present Continuous? Please check the Tense

LL94-95 –“allows for example for the construction” – Please check te word order.

L164 – “overexpress” overexpresses?

Please check the manuscript extensively for similar incomplete clauses.

We have corrected the manuscript according to the reviewer’s request.

We thank the reviewer #3 for his/her thorough examination of our study, and for helping us improve its statistical presentation.  

Round 2

Reviewer 3 Report

Comments and Suggestions for Authors

The several critical issues still must be resolved by the authors.

[1]

The authors are too bold while interpreting performance indicators without taking into account CI95s calculated.

A single example:

In case of the REMBRANDT cohort’s AUC CI95 confidence interval, there is a 95% chance that the 0.56-1.00 range contains the true AUC value, with the certain place of the true AUC undefined.

As a result, it is unreliable to state for the REMBRANDT/validation cohort with Sensitivity CI95 0.42-1.00, Specificity CI95 0.42-1.00 and AUC CI95 0.56-1.00 that “the developed logistic regression model had good performance”.

Thus, it is critical to provide the correct interpretation for the results gained.

[1a]

Please improve the statement in the abstract of the manuscript [LL33-34] on the performance of the developed model assessed via the REMBRANDT cohort.

Instead of “good”, please use “less obvious”, or “promising”, or “possible”, or smth else.

For example, “A built logistic regression model demonstrated good performance in TCGA and less obvious performance in REMBRANDT/validation cohorts” or smth else.

[1b]

In the “Results” section, please correct statements on the performance of the developed model assessed via the REMBRANDT cohort [LL204-206]

[1c]

In the “Results” section, [LL207-208] 

“Overall, we concluded that the Radscore has good performance in predicting the hypercoagulable status in LGG/GBM” – better to say “acceptable performance” or smth like this. Please improve.

[1d]

All possible reasons for the wide CI95s gained while estimating model performance using the REMBRANDT/validation cohort must be discussed in the relevant part of the Discussion section.

In particular, please mention the low number of cases in the REMBRANDT cohort as a reason for wide CI95s gained.

[1e]

In the answers for the initial review, the authors have reasoned for not performing augmentation procedures, including the REMBRANDT cohort.

To prevent readers puzzling, please include the reasons of not using the dataset augmentation procedures into the relevant subsection of the “Discussion” section.

[2]

The data reporting still must be improved extensively, for the sake of future readers.

!!!

Please understand that wide uncertainty intervals are not a problem of the study, but only a limitation produced due to the small sizes of train and test cohorts, that must be reported in a proper way for future readers.

In particular, the information is critical for understanding the sample sizes required for the future studies planned by readers and other scientific groups.

To ensure proper data reporting, several critical issues still must be resolved.

[2a]

On the Fig. 3 panels, the survival analysis lines must be presented with true CI95 (or CI90, or other) confidence bands. The visualized type of bands must be indicated in the captions for the figure.

The code for such an extension of the analysis is available at

http://www.sthda.com/english/wiki/survival-analysis-basics

or at
https://www.emilyzabor.com/tutorials/survival_analysis_in_r_tutorial.html

or at other sources.

For the help of interpretations, please refer to Coviello (https://www.graphpad.com/support/faq/confidence-intervals-vs-confidence-bands-for-survival-curves/) or other relevant source of information.

[2b]

Similar, the ROC curves depicted on the Fig 2C panels must be provided with the corresponding uncertainty bands estimated via cross-validation or another relevant technique.

The code for such an extension of the analysis is available at

https://scikit-learn.org/1.1/auto_examples/model_selection/plot_roc_crossval.html

or at other source.

The visualized type of bands must be indicated in the captions for the figure.

[2c]

The depiction of the ROC curves on the Fig S2 must be improved in a similar way.

The code for such an extension of the analysis is available at

https://scikit-learn.org/1.1/auto_examples/model_selection/plot_roc_crossval.html

or at other source.

The visualized type of bands must be indicated in the captions for the figure.

[2d]

Again, “losting the statistical significance while performing RxC analysis” is not critical, since the p-values are not the ultimate indicators of the scientific soundness of the manuscript.

Instead, it is critical to provide the readers with the real picture of the analyses performed.

In the supplement, please provide the readers with the full Cybersortx results analysed via Chi-squared test or Fischer Exact test for 2xN contingency tables with the CI95 ranges identified and visualised. Such testing might help to reveal disproportions in the sampled cell numbers for specific cell types.

To do so, please see online freeware “Epitools” (https://epitools.ausvet.com.au/chisq), which might depict CI95 intervals for cell numbers, or another statistical software/freeware capable to complete such a task.

The estimation of proportions for sampled cell numbers must be done for all types of cells simultaneously for a RxC matrix. Please do not use 2x2 designs! Please use RxC design.

[2e]

For the readers, it is critical to understand possible demographic and clinical biases inherent for datasets assessed in this study.

The Table 1 is conventional and must be retained within the main manuscript.

To ensure clear comprehension of the demographic characteristics, the age as a quantitative characteristic must be additionally visualized in the Supplement (or near the Table 1 in the main text) as Raincloud plots with jittering, the Median, IQR boxes and CI95 intervals visualized between control and affected samples.

The differences between groups must be assessed statistically.

To achieve this, it is possible to use the freeware JASP (https://jasp-stats.org/) or code (https://datavizpyr.com/grouped-violinplot-with-jittered-data-points-in-r/) or another freeware/software/code capable of drawing Raincloud plots with jittering.

For example, please see Fig.2 at Bommakanti et al. Comparative Transcriptomic Analysis of Archival Human Vestibular Schwannoma Tissue from Patients with and without Tinnitus. J Clin Med. 2023;12(7):2642. https://doi.org/10.3390/jcm12072642

Liu, Liu. Aided Diagnosis Model Based on Deep Learning for Glioblastoma, Solitary Brain Metastases, and Primary Central Nervous System Lymphoma with Multi-Modal MRI. Biology (Basel). 2024;13(2):99. https://doi.org/10.3390/biology13020099

Minor issues

[3]

The authors stated that “The seven features that were retained reflect tumour morphology and tumour texture.”

Please provide this possible interpretation of features within the “3.1. Construction and validation of an MRI radiomics model” subsection of the “Results” section.

[4]

On the Figure 5D panel, the authors must report the r-score alongside with the p-value reported.

[5]

After resolving all points of this review, please provide the readers with the code sources used by the authors additionally.

Comments on the Quality of English Language

Please check for typos.

Author Response

Dear Reviewer,

We are grateful for the time that you have spent on critically reading our manuscript. Based on the instructions that we have received, we are presenting a 2nd revised version that addresses all your comments (changes are highlighted in blue). Please see below our responses to each point that you raised.

Yours sincerely,

Antoine Galmiche

[1] The authors are too bold while interpreting performance indicators without taking into account CI95s calculated.

A single example: In case of the REMBRANDT cohort’s AUC CI95 confidence interval, there is a 95% chance that the 0.56-1.00 range contains the true AUC value, with the certain place of the true AUC undefined.

As a result, it is unreliable to state for the REMBRANDT/validation cohort with Sensitivity CI95 0.42-1.00, Specificity CI95 0.42-1.00 and AUC CI95 0.56-1.00 that “the developed logistic regression model had good performance”.

Thus, it is critical to provide the correct interpretation for the results gained.

In the 2nd revised version of the manuscript, we have removed all sentences that the reviewer perceived as being overstatements regarding the performance of the Radscore. Please see the detailed changes below.

[1a] Please improve the statement in the abstract of the manuscript [LL33-34] on the performance of the developed model assessed via the REMBRANDT cohort.

Instead of “good”, please use “less obvious”, or “promising”, or “possible”, or smth else.

For example, “A built logistic regression model demonstrated good performance in TCGA and less obvious performance in REMBRANDT/validation cohorts” or smth else.

 We have removed any statements regarding the performance of the Radscore in the revised abstract. We now give the values of AUC + IC95 without any appraisal.

[1b] In the “Results” section, please correct statements on the performance of the developed model assessed via the REMBRANDT cohort [LL204-206]

Here again, we have removed the corresponding statement. We now only mention the values of the AUC with IC95 in the two cohorts. 

[1c] In the “Results” section, [LL207-208] 

“Overall, we concluded that the Radscore has good performance in predicting the hypercoagulable status in LGG/GBM” – better to say “acceptable performance” or smth like this. Please improve.

We have replaced the statement « good » with « acceptable », as requested.

[1d] All possible reasons for the wide CI95s gained while estimating model performance using the REMBRANDT/validation cohort must be discussed in the relevant part of the Discussion section.

In particular, please mention the low number of cases in the REMBRANDT cohort as a reason for wide CI95s gained.

We have expanded this part of the discussion to mention the small number of patients in the REMBRANDT cohort (p.10, line 322 of the revised manuscript). We directly mention that this small number of patients explains the wide CI95 for the validation of the Radscore in REMBRANDT.  

[1e] In the answers for the initial review, the authors have reasoned for not performing augmentation procedures, including the REMBRANDT cohort.

To prevent readers puzzling, please include the reasons of not using the dataset augmentation procedures into the relevant subsection of the “Discussion” section.

We now mention that the use of augmentation procedures would lead to a high risk of introducing bias, given the very small number of patients (p.10, line 327).

[2] The data reporting still must be improved extensively, for the sake of future readers.

!!!

Please understand that wide uncertainty intervals are not a problem of the study, but only a limitation produced due to the small sizes of train and test cohorts, that must be reported in a proper way for future readers.

In particular, the information is critical for understanding the sample sizes required for the future studies planned by readers and other scientific groups.

To ensure proper data reporting, several critical issues still must be resolved.

[2a] On the Fig. 3 panels, the survival analysis lines must be presented with true CI95 (or CI90, or other) confidence bands. The visualized type of bands must be indicated in the captions for the figure.

The code for such an extension of the analysis is available at

http://www.sthda.com/english/wiki/survival-analysis-basics or at https://www.emilyzabor.com/tutorials/survival_analysis_in_r_tutorial.html, or at other sources.

For the help of interpretations, please refer to Coviello (https://www.graphpad.com/support/faq/confidence-intervals-vs-confidence-bands-for-survival-curves/) or other relevant source of information.

The survival analysis in Fig. 3 now include CI80 confidence bands, as described in the figure legend (line 240).

 [2b] Similar, the ROC curves depicted on the Fig 2C panels must be provided with the corresponding uncertainty bands estimated via cross-validation or another relevant technique.

The code for such an extension of the analysis is available at

https://scikit-learn.org/1.1/auto_examples/model_selection/plot_roc_crossval.html

or at other source.

The visualized type of bands must be indicated in the captions for the figure.

 The ROC analyses in Fig. 2C now include CI95 confidence bands, as described in the figure legend (line 204).

[2c] The depiction of the ROC curves on the Fig S2 must be improved in a similar way.

The code for such an extension of the analysis is available at

https://scikit-learn.org/1.1/auto_examples/model_selection/plot_roc_crossval.html

or at other source.

The visualized type of bands must be indicated in the captions for the figure.

The ROC analyses presented in the former Fig. S2 now include CI95 confidence bands, as described in the figure legend. Please note that this figure is now Fig S4.

[2d] Again, “losting the statistical significance while performing RxC analysis” is not critical, since the p-values are not the ultimate indicators of the scientific soundness of the manuscript.

Instead, it is critical to provide the readers with the real picture of the analyses performed.

In the supplement, please provide the readers with the full Cybersortx results analysed via Chi-squared test or Fischer Exact test for 2xN contingency tables with the CI95 ranges identified and visualised. Such testing might help to reveal disproportions in the sampled cell numbers for specific cell types.

To do so, please see online freeware “Epitools” (https://epitools.ausvet.com.au/chisq), which might depict CI95 intervals for cell numbers, or another statistical software/freeware capable to complete such a task.

The estimation of proportions for sampled cell numbers must be done for all types of cells simultaneously for a RxC matrix. Please do not use 2x2 designs! Please use RxC design.

We agree with the reviewer that a p value under 0.05 should not be taken as an ultimate sign of scientific validity, and we appreciate the opportunity that is given to us to present again the details of our analysis of the composition of the immune infiltrate in tumors stratified according to the Radscore.

For clarity, we now present both types of Cibersortx results: i) relative fractions, ii) absolute scores. A new figure S6 presents the conventional graphical output for these analyses.

In the 2nd revision of the manuscript, we now mention that a Chi2 analysis, used to compare the relative fractions, did not reach significance (page 9, line 282). We then present the analysis of the absolute scores. For each cell type, the new Table S4 lists the average scores with their CI95. These values being absolute scores (and not fractions), we applied Student’s t test with FDR correction. We believe that this is an appropriate way to compare these values (please note for example that other investigators use Student’s t test in high profile publications : PMID: 35584864).

In the initial version of the manuscript, we focused our attention on M0 macrophages because they present the highest fold increase in relative fraction / absolute scores in Radscorehigh compared to Radscorelow tumors. In this 2nd revision, we took the opportunity to present a more systematic and neutral presentation of the results, including the higher infiltration levels of Tregs. The Fig. 5D panel was therefore expanded to include Tregs along with M0 macrophages. We hope that the revised and more detailed presentation of our analyses will satisfy the reviewer. 

[2e] For the readers, it is critical to understand possible demographic and clinical biases inherent for datasets assessed in this study.

The Table 1 is conventional and must be retained within the main manuscript.

To ensure clear comprehension of the demographic characteristics, the age as a quantitative characteristic must be additionally visualized in the Supplement (or near the Table 1 in the main text) as Raincloud plots with jittering, the Median, IQR boxes and CI95 intervals visualized between control and affected samples.

The differences between groups must be assessed statistically.

To achieve this, it is possible to use the freeware JASP (https://jasp-stats.org/) or code (https://datavizpyr.com/grouped-violinplot-with-jittered-data-points-in-r/) or another freeware/software/code capable of drawing Raincloud plots with jittering.

For example, please see Fig.2 at Bommakanti et al. Comparative Transcriptomic Analysis of Archival Human Vestibular Schwannoma Tissue from Patients with and without Tinnitus. J Clin Med. 2023;12(7):2642. https://doi.org/10.3390/jcm12072642

Liu, Liu. Aided Diagnosis Model Based on Deep Learning for Glioblastoma, Solitary Brain Metastases, and Primary Central Nervous System Lymphoma with Multi-Modal MRI. Biology (Basel). 2024;13(2):99. https://doi.org/10.3390/biology13020099

Thank you for clarifying this request. The Table 1 was left in the manuscript as suggested, and we included a new Fig. S1 with a graphical presentation and statistical comparison of the main characteristics of the two cohorts, TCGA and REMBRANDT.

The corresponding Fig. S1 is mentioned on p.3, line 14 of the revised manuscript, where we state that « no significant differences were apparent upon a direct comparison of the basic clinical characteristics of the two cohorts, as shown in Fig. S1». We agree that this is a useful addition to the manuscript.  

Minor issues

[3] The authors stated that “The seven features that were retained reflect tumour morphology and tumour texture.”

Please provide this possible interpretation of features within the “3.1. Construction and validation of an MRI radiomics model” subsection of the “Results” section.

The statement was introduced on p.5 line 189 of the revised manuscript.

[4] On the Figure 5D panel, the authors must report the r-score alongside with the p-value reported.

This is now done as requested.

[5] After resolving all points of this review, please provide the readers with the code sources used by the authors additionally.

The R codes that were initially listed at the end of the « Supplementary Materials and methods » have been expanded.  To facilitate locating these codes, we now propose a separate text file with the source codes.

Round 3

Reviewer 3 Report

Comments and Suggestions for Authors

The authors improved the manuscript greatly.

Only minimal issues are remained to be resolved.

Minor issues

[1]

On the C and D  panels of the Fig.2 (under the AUC values written), please also indicate ns (n = 136 and n =39) characteristic for cohorts studied.

Consider to improve the size of the font.

[2]

These C and D panels of the Fig.2 might be used as a graphical abstract for the manuscript.

Consideration
(not for this manuscript)

[3]

Please also consider to read the Greenacre (2021) and original works by Aitchison to understand the compositional data phenomenon, which must be taken into account in the future studies of cell samples, expression levels and cohort stratifications.

Greenacre Michael.

Compositional Data Analysis.

Annual Review of Statistics and Its Application 2021. 8:271-99

https://doi.org/10.1146/annurev-statistics-042720-124436

Comments on the Quality of English Language

Please check the text for typos remained.

Author Response

Dear Reviewer,

Based on the instructions that we have received, we are presenting a 3rd revised version that addresses your comments. Please see below our responses to the few points that you raised.

Yours sincerely,

Antoine Galmiche

[1] On the C and D panels of the Fig.2 (under the AUC values written), please also indicate ns (n = 136 and n =39) characteristic for cohorts studied.

 Consider to improve the size of the font.

We have increased the size of the font from Arial 8 to Arial 10 for the x and y axes. The numbers are indicated as requested. For uniformity, we have also introduced these changes in Fig. S4.

[2] These C and D panels of the Fig.2 might be used as a graphical abstract for the manuscript.

We are now proposing the indicated figure panels as a graphical abstract.

Consideration (not for this manuscript)

 [3] Please also consider to read the Greenacre (2021) and original works by Aitchison to understand the compositional data phenomenon, which must be taken into account in the future studies of cell samples, expression levels and cohort stratifications.

Greenacre Michael. Compositional Data Analysis. Annual Review of Statistics and Its Application 2021. 8:271-99. https://doi.org/10.1146/annurev-statistics-042720-124436

We appreciate the suggestion of Reviewer #3 and we will carefully examine this reference for our future studies.

Comments on the Quality of English Language : Please check the text for typos remained.

 The manuscript has been carefully spell checked.